# CAUSAL CURIOSITY: RL AGENTS DISCOVERING SELF-SUPERVISED EXPERIMENTS FOR CAUSAL REPRESENTATION LEARNING

## ABSTRACT

Humans show an innate ability to learn the regularities of the world through interaction. By performing experiments in our environment, we are able to discern the causal factors of variation and infer how they affect the dynamics of our world. Analogously, here we attempt to equip reinforcement learning agents with the ability to perform experiments that facilitate a categorization of the rolled-out trajectories, and to subsequently infer the causal factors of the environment in a hierarchical manner. We introduce a novel intrinsic reward, called causal curiosity, and show that it allows our agents to learn optimal sequences of actions, and to discover causal factors in the dynamics. The learned behavior allows the agent to infer a binary quantized representation for the ground-truth causal factors in every environment. Additionally, we find that these experimental behaviors are semantically meaningful (e.g., to differentiate between heavy and light blocks, our agents learn to lift them), and are learnt in a self-supervised manner with approximately 2.5 times less data than conventional supervised planners. We show that these behaviors can be re-purposed and fine-tuned (e.g., from lifting to pushing or other downstream tasks). Finally, we show that the knowledge of causal factor representations aids zero-shot learning for more complex tasks.

## 1 INTRODUCTION

Discovering causation in environments an agent might encounter remains an open and challenging problem for causal reinforcement learning (Schölkopf (2015), Bengio et al. (2013), Schölkopf (2019)). Most approaches take the form of BAMDPs (Bayes Adaptive Markov Decision Processes) (Zintgraf et al. (2019)) or Hi-Param MDP (Hidden Parameter MDPs) (Doshi-Velez & Konidaris (2016); Yao et al. (2018); Killian et al. (2017); Perez et al. (2020)) which condition the transition $p(s_{t+1}|s_t, a_t; H)$ and/or reward function $R(r_{t+1}|s_t, a_t, s_{t+1}; H)$ of each environment on hidden parameters (also referred to as causal factors in some of the above studies). Let $s \in \mathcal{S}$, $a \in \mathcal{A}$, $r \in \mathcal{R}$, $H \in \mathcal{H}$ where $\mathcal{S}$, $\mathcal{A}$, $\mathcal{R}$, and $\mathcal{H}$ are the set of states, actions, rewards and feasible hidden parameters. In the physical world and in the case of mechanical systems, examples of the parameter $h_j \in \mathcal{H}$ include gravity, coefficients of friction, masses and sizes of objects. Typically, $H$ is treated as a latent variable for which an embedding is learned during training, using variational methods (Kingma et al. (2014); Ilse et al. (2019)). Let $s_{0:T}$ be the entire state trajectory of length $T$. Similarly, $a_{0:T}$ is the sequence of actions applied during that trajectory by the agent that results in $s_{0:T}$. In an environment parameterized by these causal factors, these latent variable approaches define a probability distribution over the entire sequence of (rewards, states, actions) conditioned on a latent $z$ as $p(r_{0:T}, s_{0:T}, a_{0:T-1}; z)$ that factorizes as

$$\prod_{i=1}^{T-1} p(r_{t+1}|s_t, a_t, s_{t+1}, z)p(s_{t+1}|s_t, a_t, z)p(a_t|s_t, z) \tag{1}$$

due to the Markov assumption. At test time, the agent infers the causal factor associated with its environment by observing the trajectories produced by its initial actions that can be issued by any policy such as model-based reinforcement learning.

In practice, however, discovering causal factors in a physical environment is prone to various challenges that are caused by the disjointed nature of the influence of these factors on the produced

trajectories. More specifically, at each time step, the transition function is affected by a subset of global causal factors. This subset is implicitly defined on the basis of the current state and the action taken. For example, if a body in an environment loses contact with the ground, the coefficient of friction between the body and the ground no longer affects the outcome of any action that is taken. Likewise, the outcome of an upward force applied by the agent to a body on the ground is unaffected by the friction coefficient. We can therefore take advantage of this natural discontinuity to discern causal factors.

Without knowledge of how independent causal mechanisms affect the outcome of a particular action in a given state in an environment, it becomes impossible for the agent to conclude where the variation it encountered came from. Unsurprisingly, Hi-Param and BAMDP approaches fail to learn a disentangled embedding for the causal factors, making their behaviors uninterpretable (Perez et al. (2020)). For example, if, in an environment, a body remains stationary under a particular force, the Hi-Param or BAMDP agent may apply a higher force to achieve its goal of perhaps moving the body, but will be unable to conclude whether the "un-movability" was caused by high friction or high mass of the body. Additionally, these approaches require human-supervised reward engineering, making it difficult to apply them outside of the simulated environments they are tested in.

Our goal is, instead of focusing on maximizing reward for some particular task, to allow agents to discover causal processes through exploratory interaction. During training, our agents discover self-supervised experimental behaviors which they apply to a set of training environments. These behaviors allow them to learn about the various causal mechanisms that govern the transitions in each environment. During inference in a novel environment, they perform these discovered behaviors sequentially and use the outcome of each behavior to infer the embedding for a single causal factor (Figure 1).

The main challenge while learning a disentangled representation for the causal factors of the world is that several causal factors may affect the outcome of behaviors in each environment. For example, when pushing a body on the ground, the outcome, i.e., whether the body moves, or how far the body is pushed, depends on several factors, e.g., mass, shape and size, frictional coefficients, etc. However, if, instead of pushing on the ground, the agent executes a perfect grasp-and-lift behavior, only mass will affect whether the body is lifted off the ground or not.

Thus, it is clear that not all experimental behaviors are created equal and that the outcomes of some behaviors are caused by fewer causal factors than others. Our agents learn these behaviors without supervision using *causal curiosity*, an intrinsic reward. The outcome of a single such experimental behavior is then used to infer a binary quantized embedding describing the single isolated causal factor. Even though causal factors of variation in a physical world are easily identifiable to humans, a concrete definition is required to back up our proposed method. We conjecture that the causality of a factor of variation depends on the available actions to the agent. If the set of actions that an agent can take is very limited, there is no way for it to discern a diverse set of causal factors in the environment.

**Definition 1** (Causal factors). *Consider the POMDP ($\mathcal{O}$, $\mathcal{S}$, $\mathcal{A}$, p, r) with observation space $\mathcal{O}$, state space $\mathcal{S}$, action space $\mathcal{A}$, the transition function p, and the reward function r. Let $o_{0:T} \in \mathcal{O}^T$ denotes a trajectory of observations and T be the length of such trajectories. Let $d(\cdot, \cdot) : \mathcal{O}^T \times \mathcal{O}^T \to \mathbb{R}_+$ be a distance function defined on the space of trajectories of length T. The set $H = \{h_1, h_2, \ldots, h_k\}$ is called a set of $\epsilon$−causal factors if for every $h_j \in H$, there exists a unique sequence of actions $a_{0:T}$ that clusters the state trajectories into two sets S and S' such that*

$$\min\{d(o_{0:T}, o'_{0:T}) : o_{0:T} \in O, o'_{0:T} \in O'\} > \epsilon \tag{2}$$

*and that $h_j$ is the cause of the trajectory of states obtained i.e.,*

$$p(o_{0:T}|do(h_j = k), a_{0:T}) \neq p(o_{0:T}|do(h_j = k'), a_{0:T}) \, \forall k \neq k' \tag{3}$$

Intuitively, a factor of variation affecting a set of environments is called causal if there exists a sequence of actions available to the agent where the resultant trajectories are clustered into two or more sets (for simplicity here we assume binary clusters). This is analogous to the human ability to conclude whether objects are heavy or light, big or small. For a gentle introduction to the intuition about this definition, we refer the reader to Appendix D.

According to Def. 1, a causal factor is a parameter in the environment whose value, when intervened on (i.e. varied) over a set of values, results in trajectories of states that are divisible into disjoint

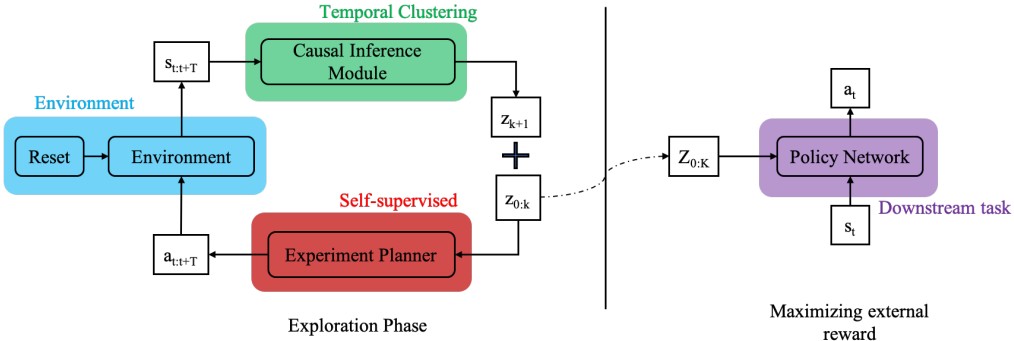

**Figure 1:** Overview of Inference. The exploration loop produces a series of $K$ experiments allowing the agent to infer the representations for $K$ causal factors. After exploration, the agent utilizes the acquired knowledge for downstream tasks.

clusters under a particular sequence of actions. These clusters represent the quantized values of the causal factor. For example, mass, which is a causal factor of a body, under an action sequence of a grasping and lifting motion, results in 2 clusters, liftable (low mass) and not liftable (high mass). However, such an action sequence is not known in advance. Therefore, discovering a causal factor in the environment boils down to finding a sequence of actions that makes the effect of that factor prominent by producing clustered trajectories for different values of that environmental factor.

Using the above, we propose an intrinsic reward, which allows our agents to discover experimental behaviors which are semantically meaningful and can be used to re-train for downstream tasks, resulting in high sample efficiency. Our work, therefore, forms an important link between structured representation learning and skill discovery, two largely disjoint fields in RL, which stand to benefit from each other.

The contributions of the work are as follows:

- We equip agents with the ability to perform experiments and behave meaningfully in a set of environments in an unsupervised manner. These behaviors can expose or obfuscate specific independent causal mechanisms that occur in the world of the agent, allowing the agent to learn about each in the absence of the others, an important human behavioral trait.

- We introduce an intrinsic reward, *causal curiosity*, which allows our agents to discover these behaviors without human-engineered rewards. The outcomes of the experiments are used to learn a disentangled quantized binary representation for the causal factors of the environment, analogous to the human ability to conclude whether objects are light/heavy, big/small etc.

- Through extensive experiments, we conclude that knowledge of the causal factors aids sample efficiency in two ways - first, that the knowledge of the causal factors aids transfer learning across multiple environments, and, second, that the experimental behaviors acquired can be repurposed for downstream tasks.

## 2 METHOD

Consider a set of $N$ environments $\mathcal{E}$ with $e^{(i)} \in \mathcal{E}$ where $e^{(i)}$ denotes the $i^{th}$ environment.

The letter $H$ is overloaded. While $H$ is a set of global causal factors (as defined in Def. 1) such that $h_j \in H$, each causal factor $h_j$ is itself a random variable which assumes a particular value for every instantiation of an environment. Thus every environment $e^{(i)}$ is represented by a set of causal factors $\{h_j^{(i)} \forall j\}$. For each environment $e^{(i)}$, $(z_{(0)}^{(i)}, z_{(1)}^{(i)}...z_{(K-1)}^{(i)})$ represents the disentangled embedding vector, such that $z_{(j)}^{(i)}$ encodes $h_j^{(i)}$.

---

**Algorithm 1** Training Scheme

---

1: Initialize $j = 0$
2: Initialize training environment set $Envs$
3: **for** iteration m to M **do**            ▷ **Experiment Planner Training Loop**
4:     Sample experimental behavior $a_{0:T} \sim \text{CEM}(\cdot)$
5:     **for** $i^{th}$ env in $Envs$ **do**
6:         Apply $a_{0:T}$ to env
7:         Collect $S^{(}i) = O_{0:T}^{(i)}$
8:         Reset env
9:         Calculate $-L(S|M)$ given that $M$ is bimodal clustering model            ▷ **Calculate Curiosity**
10:        Update CEM$(\cdot)$ distribution with highest reward trajectories
11: Use learnt $q_M(z|S)$ for cluster assignment of each env in $Envs$ i.e. $z_j^{(i)} = q_M(z|S^{(i)})$
12: Update $j = j + 1$
13: Repeat from step 2, first setting $Envs = \{e^{(i)} : z_{j-1}^{(i)} = 0\}$ and then, setting
    $Envs = \{e^{(i)} : z_{j-1}^{(i)} = 1\}$

---

## 2.1 TRAINING THE EXPERIMENT PLANNER

To learn about causal processes through interaction, the agent must produce a sequence of actions $a_{0:T-1}$ that we call *experimental behavior*, which, when applied to environment $e^{(i)} \in \mathcal{E}$, produces a sequence of observations (state) $s^{(i)} = [o_0^{(i)}, o_1^{(i)}..o_T^{(i)}]$, which is then used to infer the value of the embedding for a single causal factor $z_{(j)}^{(i)}$.

We motivate this using model selection criterion. Normally in model selection applications, the observations are fixed and the goal is to find a model $M^*$ that is closest to reality, as represented by:

$$M^* = \arg\min_M (L(M) + L(S|M)) \tag{4}$$

where $L(\cdot)$ is the description length. However, here, the situation is reversed. A simple bi-modal clustering model is fixed, motivated by Definition 1. Then, the agent is motivated to produce actions that result in observations that are best explained by this model. These discovered action sequences are the *experimental behaviors* we desire.

$$a_{0:T}^* = \arg\min_{a_{0:T}} (L(M) + L(S|M)) \tag{5}$$

where each observed trajectory $S = S(a_{0:T})$ is a function of the action sequence. As mentioned earlier, the model is fixed in this formulation; hence, the first term $L(M)$ is constant and not a function of the actions. $-L(S|M)$ that is fed back to the RL agent as a reward function to maximize. We regard this reward function as *causal curiosity*.

Note that since each causal factor has its own independent causal mechanism that causes $S$, the MDL of $S$ will be higher if multiple causal factors cause $S$. On the contrary, if the agent produces actions which result in an $S$ that is easily explained by a low-capacity bi-modal model $M$, then it will imply that $S$ is caused by fewer causal factors. Consequently, the causal curiosity reward for such an action sequence, $-L(S|M)$, will be high. Therefore, causal curiosity favors experimental behaviors that result in observations caused by few causal factors - thereby allowing us to use $S$ to infer a representation for a single causal factor. For details, please refer Appendix A.

## 2.2 CAUSAL INFERENCE MODULE

By maximizing the causal curiosity reward it is possible to achieve behaviors which result in trajectories of states only caused by a single hidden parameter. However, we wish to use the outcome of performing these experimental behaviors in each environment to infer a representation for the causal factor isolated by the experiment in question.

We achieve this through cluster membership. After training the Model Predictive Control Planner (Camacho & Alba (2013)), we sample from an action sequence $a_{0:T}$ and apply it to each of the

training environments. The learnt clustering model $M$ is then used to infer a representation for each environment using the collected outcome $S^{(i)}$ obtained by applying $a_{0:T}$ to each environment.

$$z_j^{(i)} = q_M(z|S^{(i)}) \tag{6}$$

This corresponds to Step 11 of Algorithm (1). The representation learnt is binary in nature corresponding to the quantization of the continuous spectrum of values a causal factor takes in the training set into high and low values. Note however that a binary quantized embedding is not a necessary part of our method. A dense embedding may alternatively be learnt here similar to (Perez et al. (2020); Zintgraf et al. (2019)) using approximate variational inference. However, performing interventions on a dense embedding (Section 2.3) increases the computational complexity exponentially. Balancing space and time complexity, we report results using the quantized binary form of Equation (6). We discuss the implications of increasing the complexity of $z_j^{(i)}$ in the discussion.

### 2.3 INTERVENTIONS ON BELIEFS

Having learnt about the effects of a single causal factor of the environment we wish to learn such experimental behaviors for each of the remaining hidden parameters that may vary in an environment. To achieve this, in an ideal setting, the agent would require access to the generative mechanism of the environments it encounters. Ideally, it would hold the values of the causal factor already learnt about constant i.e. $do(h_j = constant)$, and intervene over (vary the value of) another causal factor over a set of values $K$ i.e. $do(h_j = k)$ such that $k \in K$. For example, if a human scientist were to study the effects of a causal factor, say mass of a body, she would hold the values of all causal factors constant, (interact with cubes of the same size and external texture) and vary only mass to see how it affects the outcome of specific behaviors she applies to each body.

However, in the real-world the agent does not have access to the generative mechanism of the environments it encounters, but merely has the ability to act in them. Thus, it can intervene on the representations of a causal factor of the environment i.e. $do(z_i = constant)$. For, example having learnt about gravity, the agent picks all environments it believes have low gravity, and uses them to learn about a separate causal factor say, friction.

This corresponds to Step 13 of Algorithm (1). Thus, to learn about the $j^{th}$ causal factor, we repeat steps 3 onwards on each of the clusters obtained for the $j - 1^{th}$.

$$Envs = \{e^{(i)} : z_{j-1}^{(i)} = k\}, k \in \{0, 1\} \tag{7}$$

This process continues in the form of a tree (Figure 4), where for each cluster of environments, a new experiment learns to split the cluster into 2 sub-clusters depending on the value of another hidden parameter. At level $n$, the agent produces $2^n$ experiments and inference models, having already intervened on the binary quantized representations of $n$ causal factors.

## 3 RELATED WORK

Doshi-Velez & Konidaris (2016) define a class Markov Decision Processes where transition probabilities $p(s_{t+1}|s_t, a_t; \theta)$ depend on a hidden parameter $\theta$, whose value is not observed, but its effects are felt. Killian et al. (2017) and Yao et al. (2018) utilize these Hidden Parameter MDPs (Markov Decision Processes) to enable efficient policy transfer, assuming that transition probabilities across states are a function of hidden parameters. Perez et al. (2020) relax this assumption, allowing both transition probabilities and reward functions to be functions of hidden parameters. Zintgraf et al. (2019) approach the problem from a Bayes-optimal policy standpoint, defining transition probabilities and reward functions to be dependent on a hidden parameter characteristic of the MDP in consideration. We utilize this setup to define causal factors.
Substantial attempts have been made at unsupervised disentanglement, most notably, the $\beta$-VAE Higgins et al. Burgess et al. (2018), where a combination of factored priors and the information bottleneck force disentangled representations. Kim & Mnih (2018) enforce explicit factorization of the prior without compromising on the mutual information between the data and latent variables, a shortcoming of the $\beta$-VAE. Chen et al. (2018) factor the KL divergence into a more explicit form, highlighting an improved objective function and a classifier-agnostic disentanglement metric. Locatello et al. (2018) show theoretically that unsupervised disentanglement (in the absence of inductive

biases) is impossible and highly unstable, susceptible to random seed values. They follow this up with Locatello et al. (2020) where they show, both theoretically and experimentally, that pair-wise images provide sufficient inductive bias to disentangle causal factors of variation. However, these works have been applied to supervised learning problems whereas we attempt to disentangle the effects of hidden variables in dynamical environments, a relatively untouched question.

Curiosity for robotics is not a new area of research. Schmidhuber (2006), Ngo et al. (2012), Pathak et al. (2017) describe curiosity as the motivation behind the behavior of an agent in an environment for which the outcome is unpredictable, i.e., an intrinsic reward that motivates the agent to explore the unseen portions of the state space (and subsequent transitions). While causal curiosity is an intrinsic reward, it differs from these traditional definitions of curiosity in that it motivates the agent to produce structure in the outcome of its behavior.

## 4 Experiments

Our work has 2 main thrusts - the discovered experimental behaviors and the representations obtained from the outcome of the behaviors in environments. The experimental behaviors are tied to contributions 1 and 2 in the Introduction. The causal factors allow us to achieve contribution 3 in the Introduction. We visualize these learnt behaviors and verify that they are indeed semantically meaningful and interpretable. We quantify the utility of the learned behaviors by using the behaviors as pre-training for a downstream task. In our experimental setup, we verify that these behaviors are indeed invariant to all other causal factors except one.

We visualize the representations obtained using these behaviors and verify that they are indeed the binary quantized representations for each of the ground truth causal factors that we manipulated in our experiments. Finally, we verify that the knowledge of the representation does indeed aid transfer learning and zero-shot generalizability in downstream tasks.

**Causal World** We use the Causal World Simulation (Ahmed et al. (Under submission 2020)) based on the Pybullet Physics engine to test our approach. The simulator consists of a 3-fingered robot, with 3 joints on each finger. We constrain each environment to consist of a single object that the agent can interact with. The causal factors that we manipulate for each of the objects are size, shape and mass of the blocks. The simulator allows us to capture and track the positions and velocities of each of the movable objects in an environment. While, for most experiments, the 3D position and 3D pose of the blocks is used as the state at each time step, we perform ablation studies where less information is provided to the agent.

### 4.1 Visualizing Discovered Behaviors

We would like to analyze whether the discovered experimental behaviors are human interpretable, i.e., *are the experimental behaviors discovered in each of the setups semantically meaningful?* We find that our agents learn to perform several useful behaviors without any supervision. For instance, to differentiate between objects with varying mass, we find that they acquire a perfect grasp-and-lift behavior with an upward force. In other random seed experiments, the agents learn to lift the blocks by using the wall of the environment for support. To differentiate between cubes and spheres, the agent discovers a pushing behavior which gently rolls the spheres along a horizontal direction. Qualitatively, we find that these behaviors are stable and predictable. See videos of discovered behaviors `here` (website under construction).

Concurrent with the objective they are trained on, we find that the acquired behaviors impose structure on the outcome when applied to each of the training environments. The outcome of each experimental behavior on the set of training environments results in dividing it into 2 subsets. These subsets correspond to the binary quantized values of a single factor, e.g., large or small, while being invariant to the values of other causal factors of the environments. We also perform ablation studies where instead of providing the full state vector, we provide only one coordinate (e.g., only x, y or z coordinate of the block). We find that causal curiosity results in behaviors that differentiate the environments based on outcomes along the direction provided. For example, when only the x coordinate was provided, the agent learned to evaluate mass by applying a pushing behavior along the x direction. Similarly, a lifting behavior was obtained when only the z coordinate was supplied to the curiosity module (Figure 2).

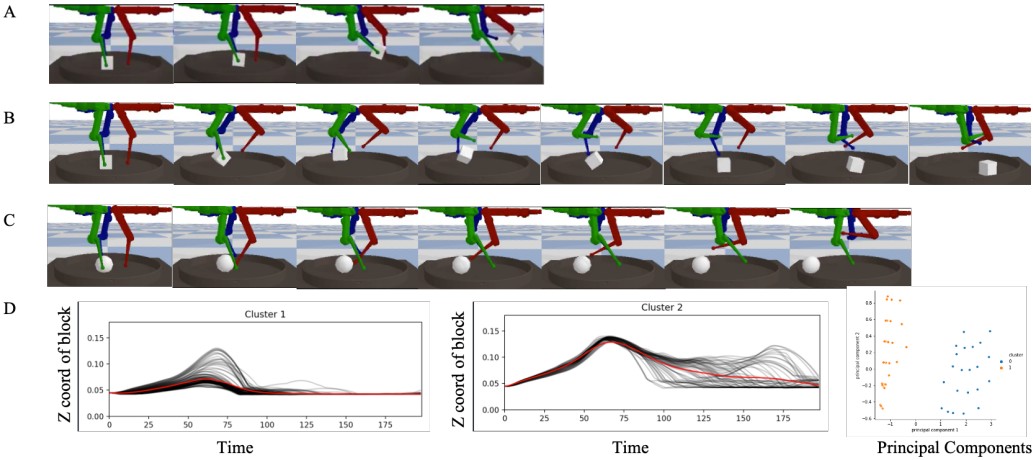

**Figure 2:** Examples of discovered behaviors. The agent discovers experimental behaviors that allow it to characterize each environmental object in a binary manner, e.g., heavy/light, big small, rollable/not rollable, etc. These behaviors are acquired without any external supervision by maximizing the causal curiosity reward. **A, B, C** correspond to self-discovered toss, lift-and-spin and roll behaviors respectively. **D** shows an ablation study, where the agent is only provided the z coordinate of the block in every environment. Each line corresponds to one environment and the z coordinate of the block is plotted with time when the discovered behavior is applied. It learns a lifting behavior, where cluster 1 represents the heavy blocks (z coordinate does not change much) and cluster 2 represents the light blocks (z increases as block is lifted and then falls when dropped and subsequently increases again when it bounces).

## 4.2 UTILITY OF LEARNED BEHAVIORS FOR DOWNSTREAM TASKS

While the behaviors acquired are semantically meaningful, we would like to quantify their utility as pre-training for downstream tasks. We analyze the performance on `Lifting` where the agent must grasp and lift a block to a predetermined height and `Travel`, where the agent must impart a velocity to the block along a predetermined direction. We re-train the learnt planner using an external reward for these tasks (Curious). We implement a baseline vanilla Cross Entropy Method optimized Model Predictive Control Planner (De Boer et al. (2005)) trained using the identical reward function and compare the rewards per trajectory during training. We also run a baseline (Additive reward) which explores whether the agent recieves both the causal curiosity reward and the external reward. We find high zero-shot generalizability and quicker convergence as compared to the vanilla CEM planner (Figure **??**). We find that maximizing the curiosity reward in addition to simultaneously maximizing external rewards results in suboptimal performance due to our formulation of the curiosity reward. To maximize curiosity, the agent must discover behaviors that divide environments into 2 clusters. Thus in the context of the experimental setups, this corresponds to acquiring a lifting/pushing behavior that allows the agent to lift/impart horizontal velocity to blocks in half of the environments, while not being able to do so in the remaining environments. However, the explicit external reward incentivizes the agent to lift/impart horizontal velocity blocks in all environments. Thus these competing objectives result in sub-par performance.

## 4.3 VISUALIZATION OF HIERARCHICAL BINARY LATENT SPACE

Our agents discover a disentangled latent space such that they are able to isolate the sources of causation of the variability they encounters in their environments. For every environment, they learn a disentangled embedding vector which describes each of the causal factors.

To show this, we use 3 separate experimental setups - `Mass`, `SizeMass` and `ShapeSizeMass` where each of the causal factors are allowed to vary over a range of discrete values. During `Mass`, the agent is allowed access to 5 environments with objects having the same shape (cuboids) and size but differing only in mass. During `SizeMass`, the agent has access to 30 environments with cuboids having sizes and masses ranging over 6 and 5 values respectively. Finally, during `ShapeSizeMass`,

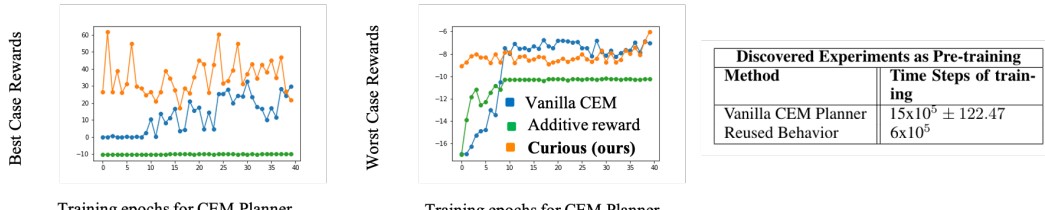

**Figure 3:** Utility of discovered behaviors. We find that the behaviors discovered by the agents while optimizing causal curiosity show high zero-shot generalizability and converge to the same performance as conventional planners for downstream tasks. We also analyze the worst case performance and find that the pre-training ensures better performance than random initialization. The table compares the time-steps of training required on an average to acquire a skill with the time steps required to learn a similar behavior using external reward. We find that the unsupervised experimental behaviors are approximately 2.5 times more sample efficient. We also find that maxizing both curiosity and external reward in our experimental setups results in sub-optimal results.

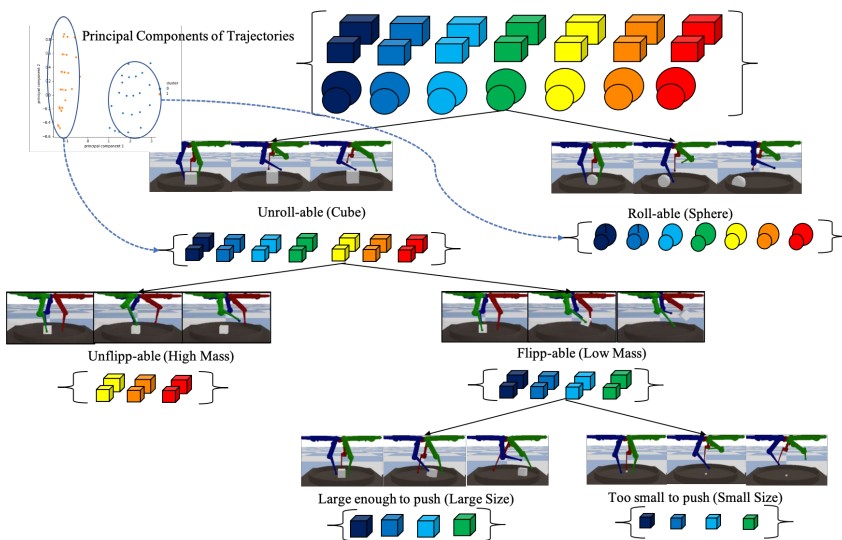

**Figure 4:** Discovered hierarchical latent space. The agent learns experiments that differentiate the full set of blocks in ShapeSizeMass into hierarchical binary clusters. At each level, the environments are divided into 2 clusters on the basis of the value of a single causal factor. We also show the principal components of the trajectories in the top left. For brevity, the full of extent of the tree is not depicted here. For each level of hierarchy $k$, there are $2^k$ number of clusters.

the agent has access to 60 environments with objects having shapes, sizes and masses ranging over 2, 6, 5, and values respectively.

During training, the agent discovers a hierarchical binary latent space (Figure 4), where each level of hierarchy corresponds to a single causal factor. The binary values at each level of hierarchy correspond to the high/low values of the causal factor in question. To our knowledge, we obtain the first interpretable latent space describing the various causal processes in the environment of an agent. This implies that it learns to quantify each physical attribute of the blocks it encounters in a completely unsupervised manner.

## 4.4 KNOWLEDGE OF CAUSAL FACTORS AIDS TRANSFER

Next, we test whether knowledge of the causal factors does indeed aid transfer and zero-shot generalizability. To this end, we supply the representations obtained by the agent during the experimental behavior phase as input to a policy network in addition to the state of the simulator, and train it for a place-and-orient downstream task (Figure 1). We define 2 experimental setups - TransferMass and TransferSizeMass. In Mass, the agent is given access to 10 environments, with 10 varying values of mass. In TransferSizeMass, the agent is allowed access to 10 environments, with 2

and 5 values of size and mass respectively. In both setups, the agent learns about the varying causal mechanisms by optimizing causal curiosity. Subsequently, using the causal representation along with the state for each environment, it is trained to maximize external reward. For details of the setup, please see Appendix B.

After training, the agents are exposed to a set of unseen test environments, where we analyze their zero-shot generalizability. These test environments consist of unseen masses and sizes and their unseen combinations. This corresponds to "Strong Generalization" as defined by Perez et al. (2020). We report results averaged over 10 random seeds.

For each setup, we train a PPO-optimized Actor-Critic Policy (referred to as **Causally-curious agent**) with access to the causal representations and a 56 dimensional state vector from the environment i.e., $a_t \sim \pi(\cdot|s_t, z_{0:K})$ (thus, a total of 57 dimensional input for `TransferMass`, and a 58 dimensional for `TransferSizeMass`). Similar to Perez et al. (2020), we implement 2 baselines - the **Generalist** and the **Specialist**. The **Specialist** consists of an agent with identical architecture as **Causally-curious agent**, but without access to causal representations (i.e., receives a 56 dimensional state vector). It is initialized randomly and is trained only on the test environments, serving as a benchmark for complexity of the test tasks. It performs poorly, indicating that the test tasks are complex. The architecture of the **Generalist** is identical to the **Specialist**. Like the **Specialist**, the **Generalist** also does not have access to the causal representations, but is trained on the same set of training environments that the Causally-curious agent is trained on. The poor performance of the generalist indicates that the tasks distribution of training and test tasks differs significantly and that memorization of behaviors does not yield good transfer. We find that causally-curious agents significantly outperform the both baselines indicating that indeed, knowledge of the causal representation does aid zero-shot generalizability.

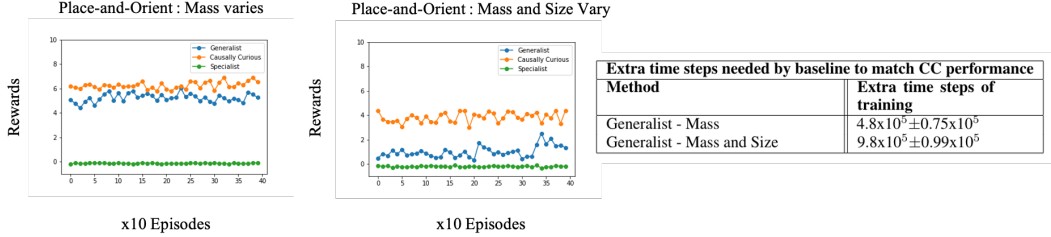

**Figure 5:** Knowledge of causal factors aids transfer. We find that knowledge of the causal representation allows agents to generalize to unseen environments with high zero-shot performance. The table depicts the extra timesteps required by the Generalist in each experimental setup to match the zero-shot performance of causally-curious agent. We find that as the number of varying causal factors increase, the difference in zero-shot performance of the Causally-curious agent and the Generalist increases, showing that the CC agents are indeed robust to multiple varying causal factors.

## 5 CONCLUSION

We introduce *causal curiosity*, an intrinsic reward that allows agents to discover binary quantized representations for the causal factors that affect environments an RL agent may encounter. We show that optimizing causal curiosity rewards results in the agent performing self-supervised experiments. We find that these experiments happen to be semantically meaningful and can be used as pre-training for downstream tasks. While our work learns binary quantized causal representations, a dense encoding may improve the amount of encoded information about the causal mechanisms of the environments. We leave this to future work.

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

## A  IMPLEMENTATION DETAILS FOR EXPERIMENT DISCOVERY

### A.1  PLANNER

The Experiment Planner consisted of a uniform distribution planner for a horizon of 6 control signals. The planner was trained using the Cross Entropy Method Model Predictive Control (Camacho & Alba (2013); De Boer et al. (2005)) on the true environment. We sampled 40 plans per iteration from the distribution initialized to uniform $\mathcal{U}(controlLow, controlHigh)$. Each of the sampled plans are applied to each of the training environments and the top 10% of the plans are used to update the distribution.

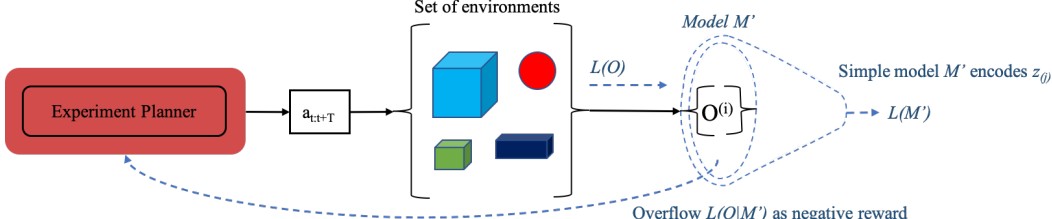

**Figure 6:** Overview of training. The experiment planner generates a trajectory of actions which is applied to each of the environments with varying causal factors namely mass, shape and size of blocks. For each environment, an observation trajectory or state $S^{(i)} \in \mathbb{S}$ is obtained. A simple model with fixed low expressive power is used to approximate the generative model for $S$. The "information overflow" $L(S|M)$ is returned as negative reward forcing $\mathbb{S}$ to be caused by few causal factors.

### A.2  TRAINING ENVIRONMENTS

The training environments vary in each experiment. In Section 4.3, we utilize 3 setups, `Mass`, `SizeMass` and `ShapeSizeMass`. For `Mass`, we allow the agent to access 5 environments with masses varying from 0.1 kg to 0.5 kg. In `SizeMass`, the agent has access to 30 environments with masses varying uniformly from 0.1 to 0.5 kg and sizes from 0.05 to 0.1 meters. Finally, in `ShapeSizeMass`, the agent has access to 60 environments, with masses varying uniformly from 0.1 to 0.5 kg, sizes from 0.05 to 0.1 meters and shapes either being cubes or spheres. During experiment discovery, in each environment, the agent has access to the position of the block in the environment along with its quaternion orientation.

The total number of causal causal factors of each environment are rather large in number due to the fact that the simulator is a complex realistic physics engine. Examples of the causal factors in the environment include gravity, friction coefficients between all on interacting surfaces, shapes, sizes and masses of blocks, control signal frequencies of the environment. However, we only vary 1 during `Mass`, 2 during `SizeMass` and 3 during `ShapeSizeMass`.

### A.3  CURIOSITY REWARD CALCULATION

We predetermine the minimum description length of the clustering model $L(M)$ by assuming that the observations $O_{0:T}$, obtained by applying experimental behavior $a_{0:T}$ are produced by a bi-modal generator distribution, where each mode corresponds to either a low or high (quantized) value of a causal factor. This also ensures that $L(M)$ is as small as possible. The planner, eq. (5) solves the following optimization problem:

$$\underset{a_{0:T} \in \mathcal{A}^T}{\arg\max}[\min\{d(o_{0:T}, o'_{0:T}) : o_{0:T} \in O, o'_{0:T} \in O'\} - \quad \max\{d(o_{0:T}, o''_{0:T}) : o''_{0:T}, o_{0:T} \in O\} -$$
$$\max\{d(o'_{0:T}, o'''_{0:T}) : o'_{0:T}, o'''_{0:T} \in O'\}]$$

(8)

the distance function $d(\cdot, \cdot)$ in the space of trajectories is set to be Soft Dynamic Time Warping (Cuturi & Blondel (2017)). The trajectory length $T$ is 6 control steps long. The objective is a modified version of the Silhouette Score (Rousseeuw (1987)).

Intuitively, Objective (8) expresses the ability of a low complexity model, assumed to be bi-modal, to encode the state $S = o_{0:T}$. If multiple causal factors control $S$, then the Minimum Description Length of $L(S)$ will be high. Subsequently, since $M$ is a simple model, the deviation of $S$ from $M$ will be high i.e. $L(S|M)$ will be high resulting in a low value of the optimization objective. $O$ and $O'$ correspond to clusters of outcomes which quantize the values of a causal factor isolated by $a_{0:T}$. $o_{0:T}, o''_{0:T} \in S$ correspond to trajectories of states i.e. observations obtained by applying $a_{0:T}$ to environments with say, low values of a causal factor while $o'_{0:T}, o'''_{0:T} \in O'$ correspond to trajectories of observations i.e. state obtained by applying $a_{0:T}$ to environments with say, high values of the same causal factor. Objective (8) attempts to ensure that these clusters are far apart from each other and are tight i.e. a simple model $M$ encodes $S$ well.

We further motivate how this formulation allows disentanglement of causal factors. A central assumption is that causal factors are independent, by definition, i.e. Independent Mechanisms Assumption Peters et al. (2017). Consider the outcome $S$ obtained by applying an action sequence $a_{0:T}$ to a set of environments. If the action sequence $a_{0:T}$ results in multiple causal factors affecting the outcome $S$, the Kolmogorov complexity of $S$ will be high. The reason for this is that each causal factor has its own independent causal mechanism (Peters et al. (2017); Parascandolo et al. (2018)) that affects $S$. Thus, given this independence, the information in $S$ will be a sum of the information "injected" into it from the multiple causes. Conversely, if the outcome $S$ obtained by applying an action sequence $a_{0:T}$ has a lower Kolmogorov Complexity, then $S$ is caused by fewer causal factors. Causal Curiosity attempts to reduce this complexity of $S$, by assuming a simple generative model $M$ is sufficient to encode $S$. Thus for experimental behaviors which allow several causes to affect $S$, the "overflow" of $S$ from $M$ will be high and subsequently the causal curiosity reward will be low. Thus, post-optimization of the objective, we arrive at an action sequence that allows for disentanglement of the causal factors.

## B  IMPLEMENTATION DETAILS FOR TRANSFER

In Section 4.4, we show the utility of learning causal representations in 2 separate experimental setups. During `TransferMass`, the agent has access to 10 environments during training, with masses ranging from 0.1 to 0.5 kg. At test time, the agent is required to perform the place-and-orient task masses 2 masses - 0.7 kg and 0.75 kg. During `TransferSizeMass`, the agent has access to 10 environments during training, with sizes from either 0.01 or 0.05 m and masses ranging from 0.1 to 0.5 kg. At test time the agent is asked to perform the task on 2 environments with masses 0.7 kg and 0.75 kg with sizes = 0.05 m.

We find that testing with large and light blocks increase the chances of accidental goal completions. Thus, during test-time, we use environments with high masses for out-of-distribution testing. The causal representation is concatenated to the state of the environment as a contextual input and supplied to a PPO-Optimized Actor-Critic Policy. The policy network consists of 2 hidden layers with 256 and 128 units respectively. The experiments are parallelized on 10 CPUs and implemented using stable baselines (Hill et al. (2018)).

The agent receives a dense reward at each time step during the maximizing external reward phase (Figure 1), the negative of the distance of the block from the goal position scaled by factor of 1000. The control signal was repeated 10 times to the actuators of the motors on each finger.

## C  IMPLEMENTATION DETAILS FOR SECTION 4.2

In section 4.2, we study how the acquired experimental behaviors obtained through Causal Curiosity can be used as pre-training for a variety of downstream tasks. The Vanilla CEM depicts the cost of training an experiment planner from scratch to maximize an external dense reward where the agent minimizes the distance between the position of a block in an environment from the goal in the Lifting setup and imparts a velocity to the block along a particular direction in the Travel setup.

$$R(a_{0:T}) = -\sum_t dist(goal_t - block_t) \tag{9}$$

The second baseline (Additive Reward) studies the setup when the agent receives both the curiosity signal and the external reward and attempts to maximize both. The agent receives access all the

training environments with varying causal factors and must simultaneously maximize both curiosity and the task reward. The equation below shows the reward maximized for the `Lifting` task.

$$R(a_{0:T}) = \sum_{envs} \sum_{t}^{T} -dist(goal_t - block_t) +$$
$$[\min\{d(o_{0:T}, o'_{0:T}) : o_{0:T} \in O, o'_{0:T} \in O'\} - \quad \max\{d(o_{0:T}, o''_{0:T}) : o''_{0:T}, o_{0:T} \in O\} -$$
$$\max\{d(o'_{0:T}, o'''_{0:T}) : o'_{0:T}, o'''_{0:T} \in O'\}]$$

(10)

The curious agent first acquired the experimental behavior by interacting with multiple environments with varying causal factors. The lifting skill was obtained during `Mass`, when the agent attempted to differentiate between multiple blocks of varying mass. The curious agent trained for 600,000 time steps on the curiosity reward. The acquired behavior was then applied to the downstream lifting task and fine tuned to external rewards. The Vanilla CEM baseline had an identical structure to that of the Curious agent, and received only external reward as in Equation (9). The additive agent simultaneously optimized both external reward and the curiosity reward as in Equation (10).

## D  INTUITION FOR DEFINITION OF CAUSAL FACTORS

We begin with a simple example of a person walking on earth. This person experiences various physical processes while interacting in her world, for example gravity, friction, wind etc. These physical processes affect the outcome of interactions of the person with her environment. For example, while jumping on earth, the human experiences gravity which affects the outcome of her jump, the fact that she falls back to the ground. Additionally, these physical processes (or causal mechanisms) are parameterized by causal factors, for example, acceleration constant due to gravity $g = 9.8m/s^2$ on earth, or coefficients of friction between her feet and the ground which assume particular numerical values.

These causal factors may vary across multiple environments. For example, the person may walk on sand or on ice, surfaces with varying frictional values. Thus the outcome of running on such surfaces will vary, running on sand will require significant effort, while running on ice may result in the person slipping. Thus the coefficient of friction between the person's feet and the surface she walks on affects the outcome of a particular behavior in said environment. In our definition, $h_j$ are causal factors such friction with some particular coefficient of friction, or gravity with acceleration constant $g$ or other. $H$ is the global set containing all such causal factors.

Now we ask the question (which we subsequently answer), given multiple environments, how would a human characterize each of them depending on the value of a causal factor? Through experimental behaviors. The human in the above example would attempt to run in each of the environments she encountered, be it on sand, on ice, in mud etc. If she slipped in an environment, she would characterize it as slippery. If she didn't, she would characterize it as non-slippery. We attempt to equip our agent with similar logic. The "sequence of actions" ($a_{0:T}$) described in our paper corresponds to the human running. The state $S^{(i)}$ in the environment $e^{(i)}$ consisting of the sequence of observations ($o_{0:T}$) corresponds to the outcome of running. $S$ might belong to either of the clusters of outcomes $S$ or $S'$ corresponding to slipping or not slipping.

## E  SCALABILITY LIMITATION

We utilize the extremely popular One-Factor-at-a-time (OFAT) general paradigm of scientific investigation, as an inspiration for our method. In the case of many hundreds of causal factors, the complexity of this method will scale exponentially. However, we believe that this would indeed be the case given a human experimenter attempting to discover the causation in any system she is studying. Learning about causation is a computationally expensive affair. We point the reader towards a wealth of material on the design of scientific experiments and more specifically the lack of scalability of OFAT (Fisher (1936); Hicks (1964); Czitrom (1999)). Nevertheless, OFAT remains the de facto standard for scientific investigation.

---

**Algorithm 2** Inference Loop

---

1: Input: Unseen Test Environment env, trained Planner and Causal Inference Module
2: Initialize $causalRep = [\ ]$
3: Initialize training environment set $Envs$
4: **for** k in range(K) **do**
5:     Reset env
6:     Sample experimental behavior $a_{0:T} \sim \text{CEM}(\cdot |\ causalRep)$
7:     Apply $a_{0:T}$ to env                        $\triangleright$ **Exploration Phase**
8:     Collect $S = o_{0:T}$
9:     Use learnt $q_M(z|S)$ for cluster assignment i.e. $z_k = q_M(z|s, causalRep)$
10:    Append $z_k$ to $causalRep$              $\triangleright$ **Causal Inference Module**
11: Learn a policy conditioned on causal factors $a_t \sim \pi(\cdot |o_t, z_{0:K})$ to maximize external reward.

---

