# OpenReview forum: "Causal Curiosity: RL Agents Discovering Self-supervised Experiments for Causal Representation Learning"
_ICLR.cc/2021/Conference — Reject_

### Official Review · AnonReviewer1 · 2020-10-28
**Great concept and appropriate framing, inadequate presentation of the technical content, missing overall set of assumptions and limitations.**

**Rating:** 5
**Confidence:** 4

**Review:**

#### **Summary**

This paper develops an intrinsic reward to help identify factors of variation within a family of MDPs. This intrinsic reward takes a form of curiosity and is used to develop initial behaviors to identify the causes of the latent variation in the environment dynamics. The experiments are used to validate the proposed intrinsic reward across several analyses used to identify its utility and effectiveness.

#### **Assessment**

I found the conceptual basis and framing of the paper to be very strong. The authors have highlighted shortcomings in prior methods that learn contextual parameters over the variation in the observed dynamics of environments and proposed an improvement in a simple combinatorial setting. I do however have strong reservations about how the paper is presented and as such cannot recommend it for publication as currently written. There is very little technical content that describe the methods and how they are used to learn effective behaviors. I also do not feel that the paper properly outlines the constraining assumptions and limitations of the proposed approach (see "Weaknesses" for more detail). The authors make claims in the abstract (e.g. "learnt... with approximately 2.5 times less data...") that aren't addressed or discussed in the main paper. In some ways this paper, as currently written, doesn't feel complete.

#### **Strengths**
- The paper appropriately criticizes prior approaches for not learning a disentangled embedding over the contextual parameters of variation within an environment. While a counterargument could be made that these approaches are looking for a low-dimensional, high information content parameterization of the factors of variation, the authors correctly point out that this limits the behaviors that could be used to efficiently identify what the causal factors are.
- The proposed solution to this shortcoming is a simple enumeration of factors of variation used to cluster trajectories with divergent behavior along the specified factor.
- This clustering of the trajectories by action sequences is used to define an intrinsic reward that encourages a clear separation between observed behaviors.
- The experiments identify salient factors of variation, allowing the learned policies to leverage the context in meaningful ways when operating in a variety of settings.

#### **Weaknesses**
I will highlight various weaknesses that I found while reading this paper below. Before I get to these things, I wanted to first discuss the major weakness of the paper that I touched on in my assessment of the paper. As currently written, there is surprisingly little technical information about how each of the components of Figure 1 are implemented. The most concerning omissions are about how the intrinsic reward is computed and how the interventions are chosen/estimated. While there are some limited details provided in the paragraphs following Equation 3, it is unclear how the clusters $S$, $S'$, $S''$ and $S'''$ are determined and how they are related. There is also no justification for the construction of Equation 3. Why 3 clustering terms? What would happen if you compared $S''$ and $S'''$? Is there any semantic meaning behind these set clusters? What is the significance of maximizing the inter-cluster distance while minimizing the intra-cluster distance? How does the objective 3 assist in computing the factors of variation $z_{(j)}^{(i)}$? What range of values are the $z_{(j)}^{(i)}$ constrained to be? How is the embedding vector provided to the policy network? Is it used to contextualize the observed state through concatenation or is it used to condition the intermediate feature representations internal to the network? What does the training and inference procedure look like? Is there psuedocode that can be incorporated into the main body of the text? The omission of these various technical details has severely weakened what appears to be a very good paper with informative results.

A second major weakness of this paper is in how the causal factors are represented. The construction of the vector $H$ assumes some *a priori* knowledge about the known causal factors of variation. While these may be possible to enumerate and abstract in simple simulated experiments such as the robotics setting demonstrated in this paper, this will not scale to larger dimension scenarios where the number of known and unknown factors of variation will potentially be very large. Without any understanding of how the individual $z_{(j)}^{(i)}$ are computed, it appears (based on the initial paragraphs of Section 2.1) that there needs to be several independent experiments + interventions for each of the K elements in the vector $z^{(i)}$. This is a significant limitation of the proposed approach in comparison with the BAMDP and HiP-MDP approaches that attempt to generalize over a dense continuous parameterization of the factors of variation. As proposed in Perez, et al [AAAI; 2020] this continuous parameterization can be factored to account for different forms of variation while also inferring a dense embedding to contextualize the dynamics and reward functions. It could be suggested that the core contribution of this paper under review is a more concrete or principled form of disentangling the causal factors of variation, but without a more formal definition it is difficult to fully ascribe where the true contributions of this paper lie. The proposed future work of determining a dense embedding seems to be overlapping with the cited work of Locatello, et al as well as the BAMDP and HiP-MDP approaches named here. Further clarification about the assumptions and limitations of the proposed approach would greatly improve the paper. As currently written, there is very little technical context by which to evaluate the core ideas and methods proposed. From a high-level view, this paper has proposed a small contribution in a highly specific sub-category of the more generalized methods is aims to improve on.

*Minor weaknesses:*
- It's not clear that the interventions will always be maximally informative. In general, what happens if more than one causal factor of variation are tied to the action. Say, for example mass and texture (eg. heavy and smooth) may jointly make it difficult for an agent to lift the object because of the confounding factor of not being able to get a secure hold on the object?
- The choice of horizon for the experiment planner (third sentence following Equation 3) seems oddly specific for the experimental environment and also seems out of place at that juncture of the paper.
- How are the "2 main thrusts" listed at the beginning of Section 4 directly linked to the proposed contributions listed at the end of Section 1?
- The statement made at the end of the first paragraph of Section 4 is possibly only true in the isolated environment that was used for the experiments. Being more forthcoming with this context and constraint is most accurate and intellectually honest.
- There are no specific details provided for the tasks used in the experiments. What are the range of shapes and sizes used? How similar are the various tasks? When evaluating for strong generalization, how far OOD are the test tasks?
- Results are said to be provided over 10 random seeds but there are no error bars or forms of variance provided in the figures.
- The only comparisons used are ablations of the proposed approach. Why not use direct baselines with the BAMDP or HiP-MDP methods? Without such comparisons, it is difficult to believe the stated conclusions that the proposed approach has fully made a contribution.
- The tables included in figures 3 and 5 are not explained nor are they very clear. What are the major takeaways that one gets from these?
- It would be nice to see where within the PCA over the trajectories, the hierarchies break down within the separate clusters.

#### **Additional Comments**
I want to close my review by commending the authors. This is good work and presents an interesting initial improvement on prior methods for generalization among related MDPs. While much of the writing and analysis of the proposed approach is qualitative in nature I feel that, if properly presented, it could be a good paper that would be worthy of acceptance to this conference. There are however significant revisions needed in order for the paper to be ready for publication. The core concepts are good and the framing is correct. But the significance and applicability of the method is hard to place due to the lack of technical detail as well as the omission of any discussion about the core assumptions and limitations of the proposed approach. Having said this, I would be inclined to raise my score if the authors were able to sufficiently address the two major weaknesses that I pointed out above and provide adequate clarification over areas of the paper that I may have misunderstood and not fairly evaluated.

- - -

### After rebuttal and discussion period

I want to state once again that it was a pleasure to iterate on this paper throughout the rebuttal period. It was gratifying to see it improve significantly. The reviewers all agree that this is a very valuable research direction and the authors have begun an extremely interesting line of inquiry. However, there is still a good deal left to be completed prior to this paper being fully suitable for publication. There are some critical areas of improvement still needed in improving the overall clarity of the proposed approach.

One piece that stood out to me when re-reading the final submission and the other discussions with the reviewers, it's left unstated with any formal language how the causal interventions are chosen. My concern along these lines are a dressing up of monte-carlo approaches of varying the separate causal factors in more sophistication than is actually present in order to make the paper seem more concrete. Not saying that this is how I read the paper, but there is some space to wonder and be concerned by this. Along these lines, the distributions $q(z|S)$ are never concretely formalized nor is the inference process fully detailed surrounding how one may infer the $z_j^{(i)}$. I could maybe guess that the $z_j^{(i)}$ could perhaps be the cluster assignment but this shouldn't be something left to speculation and guessing... A simple statement of how these distributions are parametrized or even approximated would go a long way.

As a final note, I would suggest that the authors revise their conclusion and discussion sections to incorporate the limitations inserted into the Appendix. This follows from the discussion I had with the authors about the general scoping and applicability of the proposed causal curiosity mechanism. It is implicit that the authors have robotics applications in mind yet have written in a more general purpose manner. I believe that adjusting their focus and naming the robotics-directed focus explicitly will help immensely.

In the end, I unfortunately cannot recommend this paper for publication in its current "final" form for this conference. I do however really look forward to the complete and fully published form after another series of revisions and refinements by the authors. Best wishes.

---

> ### Author Response · Authors · 2020-11-18
> **Response to Reviewer 1**
>
> Thank you for your very intelligent and constructive criticism of our work. We are grateful that you see the potential impact of the idea. We attempt to address each of your concerns below.
> 1. __Technical information about components of Figure 1__: We have added an appendix which contains details about the implementation of each component in Figure 1. The details of how the rewards were calculated and the planning algorithms used to learn the experimental behaviors can be found there. We have additionally added 2 sections in the body of the text, namely, Section 2.2 which describes how a representation for the causal factor is learnt using the outcome of applying a learnt experiment to an environment, and Section 2.3, which describes how the interventions are applied to the beliefs of the agent about the environment. We hope Section 2.2 addresses your queries about the details of the computation of the causal representation and that Section 2.3 addresses your queries with regards to the interventions required.
> The details of the transfer learning experiment have also been added to the Appendix, where we describe how the causal representation contextualizes the state of the environment. We also detail the values of sizes, masses, and shapes used at test time and during training in the appendix to confirm the out-of-distribution nature of the test environments.
> We would like to make a clarification about Equation 3 (now moved to the appendix as equation 7). We define only 2 clusters $S$ and $S'$ which refer to the quantized binary values of the causal factor in question i.e. high and low values of a causal factor, say mass. $s_{0:T},s_{0:T}'' \in S$ correspond to trajectories of states i.e. observations obtained by applying $a_{0:T}$ to environments with say, low values of a causal factor while $s_{0:T}',s_{0:T}''' \in S'$ correspond to trajectories of states i.e. observations obtained by applying $a_{0:T}$ to environments with say, high values of the same causal factor.
> Additionally, there was a typo in the equation which we have now fixed. We describe the intuitive reasoning for Equation 7 in the appendix section A.3 (last paragraph) in addition to the meaning of each of the clusters. Finally, we also add the pseudo-code for the recursion-based tree for experiment discovery in the main text. We also point the reviewer to our reply to point 6 in response to reviewer 3.
>
> 2. __Assumptions about H and related methods__:
> * __Issue of scalability__: We would like to thank the reviewer for pointing out this weakness of the method. Indeed, in the case of many hundreds of causal factors, the complexity of this method will be very high. However, we believe that would indeed be the case given a human experimenter attempting to discover the causation in any system she is studying. We utilize the extremely popular One-Factor-at-a-time (OFAT) general paradigm of scientific investigation, as an inspiration for our method. We point the reviewer towards a wealth of material on the design of scientific experiments and more specifically the lack of scalability of OFAT (Fisher, Ronald Aylmer. "Design of experiments." Br Med J 1.3923 (1936): 554-554; Hicks, Charles Robert. "Fundamental concepts in the design of experiments." (1964); Czitrom, Veronica. "One-factor-at-a-time versus designed experiments." The American Statistician 53.2 (1999): 126-131). Nevertheless, OFAT remains the de facto standard for scientific investigation (Fisher, Ronald Aylmer. "Design of experiments." Br Med J 1.3923 (1936): 554-554; Hicks, Charles Robert. "Fundamental concepts in the design of experiments." (1964)).
> * __Experiments + interventions required to learn about a causal factor__: We would like to clarify that the agent requires only a single experiment + subsequent intervention to learn about a causal factor. The reason for this is due to the formulation of causal curiosity. A central assumption is that causal factors are independent, by definition, i.e. Independent Mechanisms Assumption (Janzing, D., and Scholkopf, B. ¨ Elements of Causal Inference. MIT Press, 2017). Consider the outcome $O$ obtained by applying an action sequence $a_{0:T}$ to a set of environments. If the action sequence $a_{0:T}$ results in multiple causal factors affecting the outcome $O$, the Kolmogorov complexity of $O$ will be high. The reason for this is that each causal factor has its own independent causal mechanism (Parascandolo, Giambattista, et al. "Learning independent causal mechanisms." International Conference on Machine Learning. PMLR, 2018; Peters, J., Janzing, D., and Scholkopf, B. ¨ Elements of Causal Inference. MIT Press, 2017.) that affects $O$. Thus, given this independence, the information in $O$ will be a sum of the information “injected” into it from the multiple causes.
> (....)

---

> > ### Author Response · Authors · 2020-11-18
> > **Response to Reviewer 1 (cont.)**
> >
> > (...)
> > * Conversely, if the outcome $O$ obtained by applying an action sequence $a_{0:T}$ has a lower Kolmogorov Complexity, then $O$ is caused by fewer causal factors. Causal Curiosity attempts to reduce this complexity of $O$, by assuming a simple generative model $M$ is sufficient to encode $O$. Thus for experimental behaviors that allow several causes to affect $O$, the “overflow” of $O$ from $M$ will be high. The negative of the overflow is the curiosity reward. Thus, post-optimization of the objective, we arrive at an action sequence that allows for disentanglement requiring 1 experiment + intervention per causal factor.
> > * __Comparison to related methods__: We agree with the reviewer in that Perez et al do learn to generalize by learning a dense representation, we make the following 2 arguments:
> > __a__. Implementing Perez et al structured latent variable model requires significant hand-crafting of the representation space. The factorization of the latent space is pre-defined. Our method requires none of the above, as the agent learns about each causal factor sequentially through experiments.
> > __b__. There have been significant advances since which show that the unsupervised disentanglement by forcing factorization is insufficient to learn a disentangled latent space (most notably Locatello, Francesco, et al. "Challenging common assumptions in the unsupervised learning of disentangled representations." International Conference on Machine Learning. PMLR, 2019). Locatello et al show, both theoretically and through extensive experimentation that factorization of the latent space is insufficient for disentanglement and that some form of inductive bias is necessary. Thus, Perez at al are unable to learn a disentangled latent space. We build on this recent school of thought, learning experimental behaviors to disentangle the latent space. We also point the reviewer to the end of Section 3.1 in Perez et al where they “ leave learning how to disentangle these factors to future work”. We also hope that the additional details on how the representation is learnt, along with the pseudo-code answer the reviewer’s questions about the true contributions of our work.
> >
> > 3. __Minor weaknesses__
> > * Maximally informative experiments: While we haven’t run the exact experiment specified by the reviewer (we will shortly and report our findings), we argue that the formulation of the reward discourages multiple causal factors from affecting the outcome. We find that this is not a problem because such half-lifts or imperfect behaviors result in an increase in the complexity of the observation $O$. This will result in a lower curiosity reward and discourage such behaviors. In the example scenario offered by the reviewer, we expect that the agent will learn a pushing sequence to test out the mass of the body, followed by perhaps a grasping maneuver to test out the texture. We say this with reasonable confidence because a similar argument can be made about the experiments we have already run with the mass and size varied simultaneously - some blocks may have been too small to grasp adequately and may have slipped out while attempting a grasp-and-lift behavior. We find that this does not happen.
> > * We have moved all implementation details to the end of the paper and hope that this improves the flow of the paper.
> > * Link between experiments and contributions - we would like to point the reviewer to the added text in Section 4.
> > * We link each of the thrusts to the contributions of the paper at the beginning of Section 4.
> > * Agreed. We alter the claim in the first paragraph of Section 4.
> > * Details about transfer tasks - this has been addressed and is included in the Appendix.
> > * Variance - The values of variance have been added to the tables in Fig 3 and 5.
> > * Comparison to Hi-Param: We find it rather difficult to effectively compare the Hi-Param and BAMDP methodologies with our method due to the fact that our method contains 2 separate stages, an exploratory phase and a subsequent external reward phase, both of which have their own separate training phases. On the contrary BAMDPs and Hi-Param approaches attempt to maximize external rewards only. Our goal was to discover independent causal mechanisms and learn a disentangled latent space describing each environment while the main goal of these approaches is essentially transfer learning. We will run these as baselines for Section 4.4 before the camera-ready submission, but expect that due to the dense representation learned by such methods, they will perform better on the transfer experiment. Yet, they will not have learned about the causal mechanisms or disentangled representation on their own.
> > * Improved captions for Fig 3 and 5 - Captions of both figures have been updated to include a description of the tables. The tables link our claims of “2.5 times more sample efficient” in the abstract to the experimental results. We hope that this update addresses the reviewer’s concerns.

---

### Official Review · AnonReviewer2 · 2020-10-28
**Review of Causal Curiosity: RL Agents Discovering Self-supervised Experiments for Causal Representation Learning**

**Rating:** 6
**Confidence:** 3

**Review:**

## Summary

The authors introduce *causal curiosity*, an intrinsic reward that allows an agent to discover causal factors in an environment. The authors run experiment on `Causal World Simulation`  , a 3-fingered robot manipulating a single 3D object. These experiment show that the system:
* Learns, in a self-supervised way, interpretable behaviors corresponding to "experiments"
* Helps with two downstream tasks (`lifting` and `travel`)
* Learns an interpretable latent space that hierarchically partitions the environment in a way that isolate its causal processes
* Generalizes over unseen variations in the environment

## Analysis

I struggled to understand Definition 1. "The set $H = \\{h_1, h_2, . . . , h_k\\}$ is called a set of $\epsilon$−causal factors if for every $h_i \\in H$, there exists a sequence of actions that clusters the state trajectories into two sets $S$ and $S_0$ such that..." I can't figure out what the $h_i$ are and how "a sequence of actions" could exist for each of them.

I found interesting that the system self-discovered toss, lift-and-spin and roll behaviors, although I was not able to understand whether these behaviors allowed the agent to disentangle a specific aspect of the environment (mass, shape, size). For example, was lift-and-sping used to disentangle the mass? A better/clearer analysis of this would be helpful.

In Section 4.2, the two proposed downstream task (`lifting` and `travel`) seem very close to the behaviors learned during the self-supervised phase. As such, I'm not surprised about the fact that you're able to learn them really quickly. However, I do not find this very convincing. I'd be more impressed if the tasks were significantly different that the ones the system discovered in a self-supervised way.

In section 4.3, I struggle to understand how the hierarchical binary latent space shown in Figure 4 is discovered. Are you training first on `Mass`, then the result on `SizeMass` and then on `ShapeSizeMass`? Is Figure 4 only reporting the results after `ShapeSizeMass`? How do you build these hierarchical binary clusters? Figure 4 shows a 2D PCA with clear clusters, are you using a clustering algorithm on the 2D PCA projection of the trajectories? Did you arrive to this hierarchy through manual analysis?

Overall, this strikes me as an interesting direction to pursue, but the experiments have not fully convinced me that the proposed approach of "causal curiosity" (ie. finding a sequence of actions that maximizes the inter-cluster distance between trajectories and minimizes the intra-cluster distance) is the right approach. I believe experiments on environments that are not so clearly binary and hierarchical (shape / size / mass) would be needed.

That being said, I believe the paper is interesting enough to be accepted, especially if reviewers with more background on the topic judge it valuable.

## Typos

* I believe you are missing some $\\max\\{...\\}$ in equation 3.
* Note that *the* since each causal factor

---

> ### Author Response · Authors · 2020-11-18
> **Response to Reviewer 2**
>
> Thank you for a succinct evaluation of our work and for finding the work potentially valuable. We address each of your criticisms below.
> 1. __Explanation for Definition 1__ - We begin with a simple example of a person walking on earth. This person experiences various physical processes while interacting in her world, for example, gravity, friction, wind, etc. These physical processes affect the outcome of interactions of the person with her environment. For example, while jumping on earth, the human experiences gravity which affects the outcome of her jump, the fact that she falls back to the ground. Additionally, these physical processes (or causal mechanisms) are parameterized by causal factors, for example, acceleration constant due to gravity $g = 9.8m/s^2$ on earth or coefficients of friction between her feet and the ground which assume particular numerical values.
> These causal factors may vary across multiple environments. For example, the person may walk on sand or on ice, surfaces with varying frictional values. Thus the outcome of running on such surfaces will vary, running on sand will require significant effort, while running on ice may result in the person slipping. Thus the coefficient of friction between the person’s feet and the surface she walks on affects the outcome of a particular behavior in said environment. In our definition, $h_i$ are causal factors such as friction with some particular coefficient of friction, or gravity with acceleration constant $g$ or other. $H$ is the global set containing all such causal factors.
> Now we ask the question (which we subsequently answer), given multiple environments, how would a human characterize each of them depending on the value of a causal factor? Through experimental behaviors. The human in the above example would attempt to run in each of the environments she encountered, be it on sand, on ice, in mud, etc. If she slipped in an environment, she would characterize it as slippery. If she didn’t, she would characterize it as non-slippery. We attempt to equip our agents with similar logic. The “sequence of actions” ($a_{0:T}$) described in our paper corresponds to the human running. The observation $O^{(i)}$ in the environment $e^{(i)}$ consisting of the sequence of states ($s_{0:T}$) corresponds to the outcome of running. $O$ might belong to either of the clusters of outcomes $S$ or $S’$ corresponding to slipping or not slipping.
> For additional details of the method please see the updated manuscript in addition to our reply to reviewer 1 where we detail our inspiration from the theory of scientific investigation and design of experiments. We have added a detailed Appendix, Algorithm 1, and Sections 2.2 and 2.3. We hope that this answers the reviewer’s questions.
>
> 2. __Explanation for Section 4.2__: We would like to clarify to the reviewer that the point of Section 4.2 was to show quantitatively that the experimental behaviors discovered by our agent are semantically meaningful. We claim their meaningfulness and show anecdotes of this through our videos in Section 4.1. In 4.2 we show that these behaviors acquired through self-supervision can be reused for several semantically meaningful tasks through quantitative evaluation. Thus the relatedness. For significantly different tasks, we point the reviewer to section 4.4 where we show the utility of learning causal representations for meta-learning of downstream tasks. Section 4.2 follows a similar analysis to Section 4.2.1 in Eysenbach, Benjamin, et al. "Diversity is all you need: Learning skills without a reward function." ICLR 2019.
>
> 3. __Clarification for Section 4.3__: The agent is provided access to a full set of environments varying in mass, size and shape. The agent first learns about shape by discovering a rolling experimental behavior. Subsequently, it uses the outcome of the rolling behavior to divide the blocks into clusters on the basis of their shapes. The agent then learns experimental behaviors on each of the subsequent clusters. For brevity of Figure 4, we depict this process only on the “cube” cluster, but it learns further experiments on the “sphere” cluster also. On the “cube” cluster, the agent discovers a flipping behavior, dividing cubes into heavy and light (color-coded, warm means heavy, and cool means light). Subsequently, on the “light” cluster, the agent discovers size, by applying a horizontal pushing behavior at a predetermined height. The blocks below this height remain stationary while the larger blocks are pushed in a particular direction allowing the agent to infer the size of the block. This recursive tree traversal is detailed in Algorithm 1 now part of the main text. (...)

---

> > ### Author Response · Authors · 2020-11-18
> > **Response to Reviewer 2 (cont.)**
> >
> > 3. (...) In addition, we offer details of the implementation in the newly added appendix in addition to the details of the representation learning and interventions added in the paper. Figure 4 only reports results for `ShapeSizeMass`. The clustering algorithm is applied on the space of trajectories. To calculate Causal Curiosity to aid experiment discovery, the trajectories are first aligned to each other using time-warping and this warped representation is subsequently clustered. The PCA is reported in hindsight depicting that the learned experiment does indeed split the training environments into 2 clusters.
> > We hope that this clarifies also how the experimental behaviors identified in the videos are utilized to learn causal representations. The addition of the Section 2.2 and 2.3 along with the algorithm should also clarify this in the paper.
> >
> > 4. __Clarification on experimental setups__: We would like to clarify that the experimental setups are not binary and hierarchical. We point the reviewer to the added appendix where details of the continuous values of masses and sizes used in the experiment are detailed. The representation learned is merely binary in nature.
> > We have fixed the typos in the manuscript. Thank you for pointing them out.

---

### Official Review · AnonReviewer3 · 2020-10-28
**A compelling technique with a questionable evaluation**

**Rating:** 5
**Confidence:** 4

**Review:**

This paper considers the problem of skill discovery in settings where the data appears to be a Markov Decision Process and part of the state is unobservable. The hidden state variables are interpreted as causal factors that control important aspects of the environment  dynamics. Under this interpretation, the paper advocates for the use of a reward that encourages learned skills that exercise individual components of the hidden state. These skills are learned with a model-based RL algorithm -- one skill per causal factor -- then transferred for use in a downstream control problem that uses a different learning algorithm. The paper claims the learned skills are qualitatively meaningful, and that they enable agents to solve downstream problems without any additional training. Data used for empirical evidence comes from a simulated manipulation robot.

The paper presented several interesting ideas with the potential for high impact in robotics and reinforcement learning. I thought the paper was generally well written and the ideas well-positioned with respect to related work. I have concerns about the empirical evaluation, which I detail in several followup questions:

1. Can you explain the difference(s) in how states and observations are treated in this paper?
2. Can you verify that the structured behaviors shown in your videos are a result of the MDL reward and not the changes imposed to the observation?
3. Are the same behaviors realizable using the MDL reward as an additive bonus on the downstream problem?
4. How do the hidden causal factors numerically differ in each video?
5. How does the MDL skill differ from a skill trained from an external reward that explicitly encodes the same behavior?
6. Can you explain the notation in Equation 3? What does the minus sign denote?
7. Why is the causality formalism useful and necessary?
8. Does the MDL reward structure lead to better skills than alternatives? Examples include:  a negative linear threshold of the hidden state vector, a reward of -1 for each non-zero hidden factor, the negative magnitude of the hidden factor vector, and possible competing methods found in the literature.
9. What is the sample cost of learning an individual skill, and how does it compare to training on the downstream problem from scratch?
10. Why are the best rewards negative and the worst positive in Figure 3?
11. Over how many trials was data collected for the results in figures 3 and 5?

---

> ### Author Response · Authors · 2020-11-18
> **Response to Reviewer 3 (1/3)**
>
> Thank you for your reviews and for your questions. We appreciate that you find this work to be valuable to the research community. We attempt to answer your questions below. We have also run new experiments to answer some of your questions and details our results below in addition to updates made in the manuscript.
> 1. __Difference between state and observation__: We treat the cartesian coordinates of the block and its quaternion orientation as the state of the environment during the experiment discovery phase. The observation is merely a temporal sequence of such states obtained by applying a learnt experimental behavior to a particular environment. For a simple example providing intuition about the setup, we refer the reviewer to point 1 in our reply to reviewer 2 (which is also part of the added appendix). We hope that this answers the reviewer’s query.
> 2. __Confirmation of Utility of Causal Curiosity__: We impose no explicit changes to the observation. The agent is only provided with a set of training environments with multiple varying causal factors during training. The agent receives no additional training signal during experiment discovery. It only attempts to categorize each of the environments using the outcomes of discovered experiments. Due to the absence of any external reward, we can confirm that the behaviors discovered are due to the Causal curiosity reward. We point the reviewer to point 3 of our reply to reviewer 2 (“Clarification for Section 4.3: The agent is provided access to a full set of environments...”) for a simplified explanation for the procedure in addition to Algorithm 1 in the manuscript. The addition of Section 2.2 and 2.3 should also clarify this in the paper. Finally, we point the reviewer to point 2 b in our response to reviewer 1 (“Experiments + interventions required to learn about a causal factor: We would like to clarify that the agent requires only a single experiment...”) which should clarify how the causal curiosity reward discovers behaviors that allow the agent to learn the representation of a causal factor.
> 3. __Curiosity reward as an additive reward__: We ran an additional experiment to answer this question. We point the reviewer to the updated Figure 3 in Section 4.2. During the experiment discovery phase, we provide the agent with the training environments and supply it with both a dense reward for the tasks and the causal curiosity reward to discover experiments. We find that maximizing the curiosity reward in addition to simultaneously maximizing external rewards results in suboptimal performance. We believe this is due to our formulation of the curiosity reward. To maximize curiosity, the agent must discover behaviors that divide environments into 2 clusters. Thus in the context of the experimental setups, this corresponds to acquiring a lifting/pushing behavior that allows the agent to lift/impart horizontal velocity to blocks in half of the environments, while not being able to lift/impart horizontal velocity to blocks in the remaining environments. However, the explicit external reward incentivizes the agent to lift/impart horizontal velocity blocks in all environments. Thus these competing objectives result in sub-par performance. Due to the lack of convergence, the corresponding entry for this experiment in the adjoining table has not been included.
> 4. __Values of causal factors in experiments__: We provide details of the setup in the appendix in addition to the values of the training and test environments during the transfer experiments.
> 5. __Difference between MDL skill and skill explicit rewards__: Due to the nature of the objective, the agent must learn to perform the skill in half the environments while in the remaining environments, a separate outcome must occur. On the contrary, the skills obtained using explicit external rewards are trained in a single environment only. Thus we find that the skills obtained using Causal Curiosity are _robust_ over a set of values of the causal factor, whereas explicitly obtained skills are brittle and break down when utilized in an environment different from the one they were trained on. We also find (depicted in Section 4.2) that acquiring skills using causal curiosity is computationally cheaper than explicit learning. We show the additional time steps required to train a skill from scratch, which is evidence supporting our claims of sample efficiency.
>
> (...)

---

> > ### Author Response · Authors · 2020-11-18
> > **Response to Reviewer 3 (2/3)**
> >
> > (...)
> > 6. __Explanation for Equation 3__: We have refactored the paper in order to fit in all the requisite details and thus Equation 3 is now Equation 7 placed in the appendix. Intuitively, Equation 7 expresses the ability of a low complexity model, assumed to be bi-modal, to encode the observations $O = s_{0:T}$. A central assumption is that causal factors are independent, by definition, i.e. Independent Mechanisms Assumption (Janzing, D., and Scholkopf, B. ¨ Elements of Causal Inference. MIT Press, 2017). Consider the outcome $O$ obtained by applying an action sequence $a_{0:T}$ to a set of environments. If the action sequence $a_{0:T}$ results in multiple causal factors affecting the outcome $O$, the Kolmogorov complexity of $O$ will be high. The reason for this is that each causal factor has its own independent causal mechanism (Parascandolo, Giambattista, et al. "Learning independent causal mechanisms." International Conference on Machine Learning. PMLR, 2018; Peters, J., Janzing, D., and Scholkopf, B. ¨ Elements of Causal Inference. MIT Press, 2017.) that affects $O$. Thus, given this independence, the information in $O$ will be a sum of the information “injected” into it from the multiple causes. Conversely, if  the outcome $O$ obtained by applying an action sequence $a_{0:T}$ has a lower Kolmogorov Complexity, then $O$ is caused by fewer causal factors.
> > Equation 7 expresses the ability of $M$ to encode $O$. The first term $min$ denotes the minimum distance between trajectories with membership in separate clusters. We would like to discover an action sequence such that this distance is large. Intuitively, this means that the agent must discover an action sequence such that the outcome of applying this action sequence in environments with very different causal factors is very different e.g. applying a lifting behavior on a heavy block results in the block not being lifted, an outcome which is very different than one obtained by applying the lifting behavior to a light block i.e. the light block is lifted.
> > The negative sign on terms 2 and 3 (i.e. $max$) means that we would like to discover actions sequences such that for environments with values of the causal factor with the similar values, the outcomes of applying $a_{0:T}$ are almost the same. E.g. all light blocks should be lifted while all heavy blocks must remain stationary.
> > 7. __Utility of causality formalism__:
> > We believe Causality for RL is necessary for the following reasons.
> > * The causality formalism is necessary to efficiently learn a disentangled latent space that describes the environment an agent may encounter. Current approaches to transfer learning fail to learn a disentangled embedding for the causal factors of the environment. We detail the reasons for this in the introduction of the paper (paragraph 2 on page 2).
> > * Knowing the cause of the variation the agent encounters is an important aspect of intelligence as it enhances the robustness of its performance in a variety of environments. If the agent understands the causes for the outcome of a behavior in an environment, it is likely that the agent will be able to generalize to more diverse unseen environments. (See References) The experiments for transfer in these existing SOTA approaches in meta-RL are rather simplistic in nature. Typically, the agent receives a collection of training environments where a particular causal factor is varied. At test time the agent is asked to generalize to an unseen environment with an unseen value of the same causal factor. However, if at test-time, the value of a separate causal factor is varied (one that remained constant in each of the training environments), existing meta-RL models would fail. We posit that the reason for this is that these models have no notions of causation. If the agent were made aware of where the variation they experience is coming from, these models would not be so brittle.
> > 8. __Comparison to similar skill learning__: We would like to clarify that while the experimental behaviors obtained are similar to skill learning and that we do utilize them for downstream tasks, the main objective of causal curiosity is _causal factor discovery_. We find that existing approaches to skill do not account for this at all. To our knowledge, we propose the first unsupervised experiment discovery algorithms inspired by the One-Factor-At-a-Time (OFAT) approach in the theory Scientific Experiment Discovery. We point the reviewer for details with regards to this in our response 2a to Reviewer 1. We find that skill learning is a fortunate by-product of our approach. Comparing our method to skill learning would be unfair since no current methods in skill learning achieve causal factor discovery. For this reason we are unable to compare our approaches to existing skill learning work ("Dynamics-aware unsupervised discovery of skills.” ICLR 2020; "Diversity is all you need: Learning skills without a reward function." ICLR 2019).
> > (...)

---

> > > ### Author Response · Authors · 2020-11-18
> > > **Response to Reviewer 3 (3/3)**
> > >
> > > 8. (...) With regards to the specific formulations as suggested by the reviewer, due to the sparse representations learned by our model for the causal factors, we find that these approaches would not be directly applicable. However, we have experimented with varying forms of the reward, replacing objective 7 with silhouette score (Rousseeuw's (1987)). In other variants we removed terms 2 and 3 of Objective 7, leaving only inter-cluster distance. We also experimented with varying distance functions defined in Objective 7 - using Euclidean distance instead of Soft-DTW. We find that the current formulation is results in the most stable experimental behaviors for disentanglement.
> > > 9. __Clarification of sample cost of learning experiments__: The sample cost for learning a skill is approximately $6x10^5$ time steps of the simulator averaged over all skills acquired. In comparison, training from scratch is approximately $2.5$ times more expensive than training from scratch. We point the reviewer to the updates table in Fig 3.
> > > 10. __Typo__: This was a typo and has been fixed.
> > > 11. __Repeats__: The experiments are repeated across 10 random seeds. The figures have been updated with the values of variance.
> > >
> > > [1]. Rakelly, Kate, et al. "Efficient off-policy meta-reinforcement learning via probabilistic context variables." International conference on machine learning. 2019;
> > > [2]. Finn, Chelsea, Pieter Abbeel, and Sergey Levine. "Model-agnostic meta-learning for fast adaptation of deep networks." arXiv preprint arXiv:1703.03400 (2017);
> > > [3]. Gupta, Abhishek, et al. )"Meta-reinforcement learning of structured exploration strategies." Advances in Neural Information Processing Systems. 2018).

---

> > > > ### Comment · AnonReviewer3 · 2020-11-24
> > > > **Response to 9-11**
> > > >
> > > > 9. There are not enough experimental details provided to reproduce this result. At what point did you stop training and record the sample cost?
> > > > 11. The reported error should be a standard error or 95% confidence interval. The variance is not helpful, because it can correspond to several different confidence intervals, and it's not in the same units as the evaluation metric.

---

> > > > > ### Author Response · Authors · 2020-11-25
> > > > > **Reply 9-11**
> > > > >
> > > > > 9. The following details have been added to the newly added appendix C. The curious agent first acquired the experimental behavior by interacting with multiple environments with varying causal factors. The lifting skill was obtained during ```Mass``` when the agent attempted to differentiate between multiple blocks of varying mass. The curious agent trained for 600,000 time-steps on the curiosity reward. The acquired behavior was then applied to the downstream lifting task and fine-tuned. The Vanilla CEM baseline had an identical structure to that of the Curious agent but received only the external reward as in Equation (9). The additive agent simultaneously optimized both the external reward and the curiosity reward as in Equation 10.
> > > > > 10. Changed to standard error.

---

> > > ### Comment · AnonReviewer3 · 2020-11-24
> > > **Response to 6-8**
> > >
> > > 6. The equation has changed. So my earlier question does not apply.
> > >
> > > 7. Does the causality formalism afford any better analysis that a POMDP formalism or state aggregation formalism doesn't here? It's not clear how and if interventional analysis is being leveraged. Everything else appears to be a regular MDP with latent state components.
> > >
> > > 8. This claim seems backwards to me. Discovering latent state components is a means to learn a skill. The purpose of the proposed curiosity mechanism should be to experience more downstream reward (i.e. improve the learned downstream skill in this case). For what other reason would an RL agent need to reason about latent state components? It's reasonable then to ask how other conventional (non-curiosity driven) agents perform in comparison to this method.

---

> > > > ### Author Response · Authors · 2020-11-25
> > > > **Reply 6-8**
> > > >
> > > > 6. Noted.
> > > > 7. The main improvement over POMDP approaches is that we are now able to learn an interpretable, disentangled latent space, which is not possible using standard POMDP transfer approaches. Standard POMDP approaches indeed also assume that the state $S = O_{0:T}$, however, what accompanying action sequence produces this is never considered. We show how causal curiosity can be used to expose certain parts of the hidden state (what we refer to as causal factors) while obfuscating other parts. How the causality formalism has been implemented is now part of Section 2.2 and 2.3, where “approximate” interventions i.e. interventions on beliefs help the agent learn about various causal mechanisms.
> > > > 8. The reasoning about latent components is necessary for transferring across multiple environments with varying values of such latent components (Section 4.4). Our inspiration was to improve upon (Perez, Christian F., Felipe Petroski Such, and Theofanis Karaletsos. "Generalized Hidden Parameter MDPs Transferable Model-based RL in a Handful of Trials." arXiv preprint arXiv:2002.03072 (2020)) where an entangled latent space representation is used to transfer across multiple environments. However, disentangling the hidden state (i.e. causal factor), requires learning an action sequence that allows the agent to expose one causal mechanism and obfuscate others. The skills acquired were used to learn about the latent components to enable transfer. We will run additional experiments to compare our work to DIAYN or DADS or other skill learning paradigms before the camera-ready paper.

---

> > ### Comment · AnonReviewer3 · 2020-11-24
> > **Response to 1-5**
> >
> > Thank you for your reply.
> >
> > 1. I think there is an issue with the way states are conceptually defined. In general, they should constitute all the information necessary for predicting the future. But here the 'state' only contains a subset of that information, and the hidden factors presumably contain the remaining information (Eq 1). Most would therefore consider the 'state' that the paper uses to be the observation. Sorting these concepts out so they're logically consistent requires some care, because, without it, readers will struggle to understand the results, and struggle to differentiate prior work from what is presented.
> >
> > 2. I'm referring to the situations where you provided different observation (your 'state') vectors. "We also perform ablation studies where instead of providing the full state vector, we provide only one coordinate (e.g., only x, y or z coordinate of the block). ... For example, when only the x coordinate was provided, the agent learned to evaluate mass by applying a pushing behavior along the x direction." My question was about isolating the effect of your algorithm when the observation stream was changed. It's not clear that one needs curiosity to discover such skills, because changing the observation stream imposes an inductive bias on maximizing the reward for that component. This brings me to the next question I asked.
> >
> > 3. One could verify that indeed curiosity was pulling some extra weight if it were compared to a more conventional approach, where the MDL reward was included as a bonus, or one where the MDL reward was completely absent. It's not clear how the experiment or the reward were assigned from what is written in Section 4.2. It would be helpful to include explicit mathematical definitions of the rewards used to generate the results.
> >
> > 4. Appendix A.2 does not contain enough detail to reproduce this experiment. It needs to exactly describe what was implemented and what 'causal factors' were varied.
> >
> > 5. "we find that the skills obtained using Causal Curiosity are robust over a set of values of the causal factor, whereas explicitly obtained skills are brittle and break down when utilized in an environment different from the one they were trained on." I'm not sure what point is being made here. My question was asking about the utility of the curiosity procedure again. Why are the skills more useful than simply encoding the desired behavior in the 'external' reward?

---

> > > ### Author Response · Authors · 2020-11-25
> > > **Reply for 1-5**
> > >
> > > Thank you for your reply. Apologies if the following reply is rather hurried. I am to fly out on a long haul flight in the next few hours.
> > > 1. Thank you for pointing this out. We have changed our notation to remain consistent with the current notation POMDP notations. The agent receives observations from the environment and a sequence of such observations is treated as a state which is subsequently used to infer embeddings for the causal factors of the environment. We utilize this vocabulary to address the rest of your comments.
> > > 2. The experimental findings that we report are using the full observation as input. The ablations were simply additional experiments depicted in Figure 2. The rest of the results in the paper are using the full observation stream. Hope this clarifies this issue.
> > > 3. Those are exactly the experimental scenarios we have run. The Vanilla CEM depicts the cost of training an experiment planner from scratch to maximize an external dense reward where the agent minimizes the distance between the position of a block in an environment from the goal in the Lifting setup and imparts a velocity to the block along a particular direction in the Travel setup. This is the setup without the curiosity reward. The second baseline (Additive Reward) studies the setup when the agent receives both the curiosity signal and the external reward and attempts to maximize both. We add appendix C for the mathematical equations for transfer.
> > > 4. Appendix A.1 contains the CEM planner details, A.2 contains the details of the training environments including which causal factors were varied and over what values, A.3 contains how causal curiosity was implemented, B contains the implementation details for the transfer experiment in Section 4.4, and Appendix C contains the details of the exact rewards supplied for section 4.2. We hope this addresses your concern.
> > > 5. The skills from the curiosity reward are learned in a more sample efficient way as compared to explicitly maximizing external rewards. Section 4.2 shows this. When maximizing external rewards in a particular environment, the agent learns a skill that works in only that particular environment. In Section 4.2 for example, the agent maximizing the lifting reward can only do so for an environment with a block of mass 0.03 kg will likely fail if a heavier block is substituted, or if the size of the block is varied. On the contrary, skills discovered through curiosity are invariant to varying causal factors. Consider Figure 4, the “flipping” skill obtained is robust to size, i.e., the agent can flip both large and small blocks as long as they are light. This is the robustness we would like to highlight.

---

### Official Review · AnonReviewer4 · 2020-10-30
**Initial review of Submission 764**

**Rating:** 3
**Confidence:** 4

**Review:**

**Summary**

This paper describes the definition of an intrinsic reward designed to encourage an agent to behave in such a way that they can distinguish between MDPs that have different values of some hidden parameter that affects the way the MDP behaves. Proof-of-concept experiments are included to allow discussion of how providing this classification of MDPs to an agent that needs to learn how to perform some externally-designed task allows it to learn to complete the task faster and discussion of how the agent's observations affect its learning and behaviour.

**Strengths and Weaknesses**

The primary problem with this paper is that the description of the proposed method is too imprecise to allow me to review the quality of the research completed. I feel that the language and structure are not at an acceptable standard for ICLR. Some acute problems are that the protocol for training as well as some key algorithmic choices are never specified (i.e., the clustering algorithm and the planning algorithm for selecting action trajectories) and the notation and language around causal factors are not clear.

When I say that the protocol for training is not specified, I am referring to many missing elements, like how the training dataset is composed and what information the agent is given about each environment it is placed in (e.g., is the agent able to distinguish one environment from another?) The explanation of how the clustering tree is formed lacks substance: Do the human experimenters change the composition of the training data once one causal factor has been learned sufficiently well (and how?), or is the agent able to distinguish which node of the tree a given environment belongs to? For example, what does it mean to "perform an intervention on the embedding of the causal factor isolated by the initial optimization of causal curiosity"? This is a very important aspect of the project, so it is critical to explain in plain language what you did.

For the sake of reproducibility and understanding, it is necessary to know what algorithm was used for clustering and what algorithm was used to adapt the agent's behaviour to increase the amount of intrinsic reward received.

One strength of the paper is how the authors have included intuitive examples throughout to help explain the intentions behind their design (e.g., "For example, if a body in an environment loses contact with the ground …"), though in some cases the language applied to the examples makes it unclear how they should be understood. For example, when you say "examples of the parameter $H$ include gravity, coefficients of friction…," it sounds like each $H$ could be any one of those examples, but from Definition 1, I suspect $H$ must be a set whose elements could include all or some of the examples. This should be clarified.

**Recommendation**

I am recommending that this paper be rejected because it is not written with sufficient clarity to understand the research project being reported.

**Specific Examples of Lack of Clarity**

Definition 1 seems critical to the paper, and as such, I think it needs to be written very carefully and clearly. The relationship between H and its associated sequences of actions is very unclear to me. Does each $h_i$ require a different sequence of actions from the sequence associated with $h_j \neq h_i$? (The definition doesn't seem to have that requirement.) An explanation of what it means for a sequence of actions to cluster state trajectories also seems to be needed. Later in the paper, Definition 1 is described using the phrase "when intervened on over a set of values" and I'm really not sure what this means practically.

As a reader, I need help with understanding your choice of notation. For example, I intuitively suspect that you as authors consider $H$ (pp. 1-2) and $H^{(i)}$ (p. 3) to be related by virtue of their very similar notation, but I don't know how I am supposed to connect them in my head, other than I know that both are made up of causal factors but one is a set and the other is a vector.

The connection to Rousseeuw's (1987) work is confusing. Is the reward to the agent actually given by Equation (3), which appears to be a variation of silhouette width as defined by Rousseeuw? The presentation makes it sound like Equation (3) and Silhouette Score are essentially unrelated.

I suggest rephrasing the introduction of notation on page 1 to reduce ambiguity. I think that $z$ is a latent representation of the environment (my guess based on notation conventions), but I initially assumed that $z$ symbolically represented an environment based on the text.

It is unclear whether you are using the term "causal factor" in the same way as you have noted that previous studies do (p. 1), or whether it has some other meaning in this paper.

Based on notation, I think that a hyper-parameter $H \in \mathcal H$ (also referred to as simply a parameter) is equivalent to a hidden parameter (or causal factor) in the noted literature. This use of the term hyper-parameter is confusing and if I understand correctly, sticking with the term hidden parameter would be clearer (unless there is a reason I am missing for using hyper-parameter).

**Additional Feedback (Here to help, not necessarily part of decision assessment)**

It was unclear what the set of trajectories during a rollout would look like. In other papers, trajectory sometimes refers to any observed sequence of state-action-rewards, while in others it might refer specifically to such a sequence that begins and ends at the same time as an episode. In definition 1, the phrase "Let T be the length of the trajetcories during each rollout and $s_{0:T} \in S^T$ denotes a trajectory" (p. 2) makes me think there might be multiple trajectories per rollout, so I would like to be clear on how each is defined.

Should $r_{0:T}$ be defined explicitly on page 1 too?
I encourage you to be careful with your language when you define mathematical objects to share the name of a naive concept like causal factor. The way the definition is presented already suggests that this definition doesn't capture everything you would like it to capture because of simplifying to binary clusters. While tying it to the naive concept can help provide some intuition about your intentions, I could see this name causing some confusion when this idea develops further.

"trajetcories" (p. 2)

"simplicitly" (p. 2)

I was surprised by the curly braces used for the observations (p. 3), which I think are ordered sequences rather than sets? (Typo?)

There seems to be a typo in Equation (3): I think there should be "max" on the second and third terms.

If you need more space to provide a summary of the algorithm followed by your system, I don't think that the connection to model selection on page 4 is clearly necessary to the paper.

You'll probably want to go through the bibliography to check for capitalization errors like "Soft-dtw," "beta-vae," etc.

---------------------------------------------------------------

**UPDATE after author rebuttal**

The authors have made substantial improvements to the paper over the rebuttal period, but there is still work to be done to make this paper communicate as clearly as it should for publication. I share the concerns of Reviewer 1 and feel that this paper would benefit from more thought into how to present the ideas and another round of reviews.

**Additional Feedback for future revision**

The preamble to Section 2 is now quite helpful, but it would be even more helpful if placed prior to Definition 1! Helping the reader to think about the elements of $H$ as random variables and to think of causal factors as determining which environment you end up in would aid understanding of Definition 1.

Depending on who you want your audience to be, I suggest considering adding an explanation of the "do" notation. While the sub-community focused on causality may be aware of the notation, the community interested in intrinsic motivation more broadly would probably be interested in this paper but not know the notation.

I think that the following sentence could be quite helpful if written in a different way: " In our definition, $h_j$ are causal factors such friction with some particular coefficient of friction, or gravity with acceleration constant $g$ or other" (p. 14). It would be helpful to have the variable (e.g. amount of friction) separated from its value (which might represent the coefficient of friction). I'd like the concept that causal factors are variables that could take on any of a set of values made more explicit.

"the outcome of running" (p. 14) is slightly ambiguous (at first I read it that running was the outcome, rather than the experiment. A phrase like "the outcome of the attempt to run" or the outcome of the running experiment" might be clearer.

Consider using the singular they in your human example (p. 14). A decent primer: https://apastyle.apa.org/style-grammar-guidelines/grammar/singular-they

Additional typos found:
Let $o_{0:T} \in \mathcal O^T$ denotes → denote (p. 2).
$k$ is both the length of the set $H$ and the comparator with $h_j$ (p. 2) Maybe don't use $k$ and $K$ either, since it still gives me the vague sense that they might be related, but I'm pretty sure they're not.
Neither CEM nor MDL is written out in full (p. 4 is first use).
"as compared to the vanilla CEM planner (Figure??)." (p. 7)
"are causal factors such friction" → such as friction (p. 14)

---

> ### Author Response · Authors · 2020-11-18
> **Response to Reviewer 4**
>
> We thank you for your comments. We present the updated manuscript which we hope covers your concerns. This update contains:
> * __Revamped Methods section__, with added details about representation learning and interventions.
> * __Interventions__: The sections about interventions have been rewritten in a simpler manner.
> *  __Newly added Algorithm 1__, which details the method.
> * __Appendix__: 3 sections in the appendix detailing algorithmic choices, details of the transfer experiments and the intuition behind our approach.
>
> We attempt to address each of your concerns below.
> 1. __Details of algorithmic choices, training dataset, interventions__: We have added Appendix A to the manuscript which details the choice of the algorithm, the training scheme, and computations of reward. The appendix also contains the details of the training environments for experimental behavior discovery as well as transfer learning. We also point the reviewer to updated sections 2.2 and 2.3 which detail how the causal factor representation was obtained and how the interventions on beliefs were performed. Finally, we point the reviewer to the added Algorithm 1 in the manuscript which details the methodology. We confirm that our agent automatically learns which node in the tree an environment belongs to. The explanation for the interventions has been updated in Section 2.3. The details of the clustering algorithm and planner are provided in the appendix.
> 2. __The relationship between H and actions__: Each $h_i$ element of $H$ represents a causal factor that requires a specific sequence of actions which fleshes out the effects of that particular causal factor in the environment. We point the reviewer to the appendix C where a more intuitive example is provided. The explanation for interventions is provided in Section 2.3. Additionally, the explanation for clustering of trajectories is also added in the beginning of Section 2.1.  Definition 1 has been rewritten for clarity.
> 3. __Notation for set $H$__. We agree that $H$ is overloaded. We added a note in the first paragraph of Section 2 to elaborate more on the definition of $H$.
> 4. __Rephrasing of training objective__: An updated definition of the training objective can be found in Appendix A. We use a version of the Silhouette Score which has been detailed there along with the requisite citation. Training objective (formerly Equation 3, has been moved to the Appendix as Equation 7).
> 5. __Ambiguity about z__: We have rephrased Section 1 to remove the ambiguity of the latent variable z. We have also removed the “hyper-parameter” and stuck with “hidden parameter” or “causal factor” throughout the paper.
> 6. __Rephrasing introductory notation__: We have changed the notation in definition 1 and altogether removed rollouts to prevent confusion. There is a single trajectory per rollout i.e. application of an action sequence to a single environment.
> 7. Typos have been dealt with.

---

### Author Response · Authors · 2020-11-18
**General Comments on Updated Manuscript**

We would like to thank the reviewers for their extremely useful feedback. We are grateful that they were able to recognize the novelty and the potential value of the work to the research community at large. We are also glad that they deem this work to be of potentially high impact in robotics and reinforcement learning. We found that the most common issues with the manuscript were to do with clarity, which we have subsequently improved. We have attempted to address each of the qualms of every reviewer and find that incorporating their views has highly enriched the manuscript. We thank the reviewers, ACs, PCs, and organizers for their efforts!

---

### Decision · Program_Chairs · 2021-01-07
**Final Decision**

**Decision:**

Reject

**Comment:**

This paper discusses how one can equip reinforcement learning agents with an intrinsic reward function that helps identifying factors of variation within a family of MDPs, effectively allowing agents to do experiments in the environment. This is interpreted as causal factors that control important aspects of the environment dynamics.

Although this is a very relevant topic and there was extensive discussion during the discussion phase, with reviewers acknowledging that the final version of the submitted manuscript substantially improved over the original submission, most reviewers still recommend the rejection of the paper. This is mainly due to the assessment that there are still several unclear technical aspects related to the paper. Shortly, the reviewers felt that the paper had important clarity issues, that the claims being made were imprecise, and that there was a dearth of details about the empirical results, making them not fully convincing.

I strongly recommend the authors to take the reviewers suggestions into consideration to have a much stronger submission to future venues.